# Evidence of Better Psychological Profile in Working Population Meeting Current Physical Activity Recommendations

**DOI:** 10.3390/ijerph18178991

**Published:** 2021-08-26

**Authors:** Daniela Lucini, Eleonora Pagani, Francesco Capria, Michele Galliano, Marcello Marchese, Stefano Cribellati

**Affiliations:** 1BIOMETRA Department, University of Milan, 20129 Milan, Italy; 2Exercise Medicine Unit, Istituto Auxologico Italiano IRCCS, 20135 Milan, Italy; 3Corso Magenta 83/2, 20123 Milan, Italy; pagani.eleonora01@gmail.com; 4Assidim, 20122 Milan, Italy; francesco.capria@assidim.it (F.C.); michele.galliano@assidim.it (M.G.); Marcello.marchese@assidim.it (M.M.); 5SEGE srl, 20146 Milan, Italy; stefano.cribellati@se-ge.com

**Keywords:** lifestyle, prevention, workplace health promotion, stress, exercise

## Abstract

Workplace Health Promotion (WHP) may improve health, productivity and safety and reduce absenteeism. However, although desirable, it is difficult to design tailored (and thus effective) WHP programs, particularly in small–medium companies, which rarely have access to sufficient economic and organizational resources. In this study, 1305 employees filled out an online anonymous lifestyle questionnaire hosted on the website of a non-profit organization, which aims to promote a healthy lifestyle among workers. The data show gender differences regarding stress perception and, in the working population meeting current physical activity recommendations (threshold = 600 MET·min/week), they point out the evidence of a better psychological and nutrition profile, a perception of better job performance, and improved sleep and health quality. Moreover, a unitary index (ranging from 0–100 (with higher scores being healthier)), combining self-reported metrics for diet, exercise and stress, was significantly higher in active employees (67.51 ± 12.46 vs. 39.84 ± 18.34, *p* < 0.001). The possibility of assessing individual lifestyle in an easy, timely and cost-effective manner, offers the opportunity to collect grouped data useful to drive tailored WHP policies and to have metric to quantify results of interventions. This potentiality may help in creating effective programs and in improving employees’ and companies’ motivation and attitude towards a feasible WHP.

## 1. Introduction

Lifestyle Managing Programs may be considered a sustainable tool at individual and global levels [1]. They recommend that action be taken in the present (to foster a healthy lifestyle) may preserve a greater good (Health) which, otherwise, might disappear in future. Moreover, the prevention of chronic non-communicable diseases (thanks to healthy nutrition, physical exercise, good sleep quality, quitting smoking and stress management) represents an important tool to grant benefits at the global level, saving the economic resources that might be necessary to manage those chronic diseases. Recent data show that a healthy lifestyle, particularly being physically active [2], may reduce the risk for severe infectious diseases outcomes, such as COVID-19.

Lifestyle Managing Programs may also provide a sustainable tool for companies: by improving health, they may improve productivity and safety and reduce absenteeism [3,4,5,6]. The workplace could represent an ideal setting to implement these interventions; however, there are many policies and practice in place [7,8] and many programs are not effective [8,9,10,11,12]. The European Union defined policy direction [13] to stimulate Workplace Health Promotion (WHP), which is defined as “the combined efforts of employers, employees and society to improve the health and wellbeing of people at work” [14], adapting the WHO definition of Health Promotion: “The process of enabling people to increase control over, and to improve, their health” [15]. Nevertheless, WHP remains an issue [7] and efficacy varies considerably across the different approaches to intervention. Effective programs [8,10,11,16,17,18,19] are frequently designed by considering specific characteristics and needs, and tailoring interventions to employees’ behaviors. In this context, the assessment of employees’ lifestyle may represent a pivotal strategy [20], focusing on lifestyle behavior (physical activity, nutrition, stress, smoking, etc.), more than on traditional cardiometabolic risk factors (cholesterol levels, blood pressure, etc.). This approach suggested by many current guidelines [21,22,23,24] may be particularly suitable in the workplace setting, increasing employees’ compliance with the assessment and reducing costs. In a previous paper [25], we showed that a unitary index (which ranged from 0 to 100, with higher scores being healthier), combining self-reported metrics for diet, exercise and stress, was significantly associated with clinical and lab results and anthropometric data, predicting levels of cardiometabolic risk, and representing a potentially useful, low-cost, tool that could be employed in the working population. The possibility of assessing individual lifestyle in an easy, timely and cost-effective manner offers the opportunity to give individual advice and collect grouped data useful to drive tailored WHP company policies. Awareness about one’s own lifestyle and its relationship with own health may be considered as the first step to empower an individual to improve her/his behavior [8,18,19,20,26,27], and awareness about grouped employees’ lifestyle and its link with productivity may be considered the first step to empower a company to adopt tailored WHP policies.

Another possible barrier to the implementation of effective WHP is represented by the difficulties that small and medium companies (which actually represent a huge part of the workforce) have in funding the economical and organizational resources needed to establish WHP policies.

A particularly interesting issue in WHP is the relationship between some lifestyle components (nutrition, exercise, sleep, smoke, etc.) and stress, considering both the possible role of stress in worsening behavior [28,29], and the possible role of healthy behavioral choices, particularly physical activity, as stress management strategies [20,28,30,31,32]. Stress represents an important issue at present in the work environment [7,33,34], especially considering the important changes in work due to the COVID-19 pandemic. Many causes of stress (stressors) may not easily be modified, as they are outside individual and/or company control. On the other hand, interventions based on the proactive role of the individual/company, aiming to improve lifestyle, are considered a possible strategy to improve wellbeing, manage stress [20,28,30,31,32,35], avoid unhealthy behavior that may worsen individual health and guarantee that employees can access the health resources necessary to cope with inevitable work stressors. 

The goal of this study was to verify the ability of an anonymous, simple, on-line lifestyle questionnaire, offered also to small and medium companies located in the northern part of Italy, to reveal grouped information on employees’ lifestyle, which would be useful to tailor WHP interventions.

## 2. Materials and Methods

The study involved 1333 employees of several Italian companies who, on a voluntary basis, randomly filled out an anonymous lifestyle questionnaire, from January 2021 to April 2021, on the web page of Assidim (a non-profit association which provides the associated companies, their employees and their families, providing financial assistance and support in case of disease, accident, invalidity and death) which, since its foundation in 1981, has focused on the promotion of a healthy lifestyle among workers and associated companies. Every employee who visited the Assidim web page could fill out the questionnaire.

The questionnaire was anonymous and included a question regarding whether the subject benefited from Assidim services or not. It was designed, as previously described [25], to obtain data on lifestyle (exercise, diet habits, sleep hours, smoking, alcohol and indices of stress), working role, perception of quality of personal health, on sleep, on job performance and on presence of chronic disease. Since the questionnaire is part of a campaign aimed to motivate workers to improve behavior, we provided participants with a personalized immediate report based on the filled information.

We considered for analysis only the full completed questionnaire (*n* = 1333). A quality analysis of the collected data was conducted to eliminate non-realistic data from the dataset, and 1305 questionnaires (98%) were ultimately included in the statistical analysis. We also collected anthropometric data: weigh, height and waist circumference (WC), an important parameter which can be used to predict cardiometabolic risk [36]. Considering that people are rarely aware of their WC and need to measure it, we asked them to also include their pant size (generally well-known information). Only plausible WC data (matching with pant size) (87.3%) were considered in the statistical analysis.

Perception of quality of sleep and quality of health was assessed, providing nominal self-rated Likert scales from 0 (‘bad’) to 10 (‘very good’) for each measure. Perception of job performance was assessed, providing nominal self-rated Likert scales from 0 (‘bad’) to 5 (‘very good’).

### 2.1. Lifestyle Assessment

Physical activity (weekly physical activity volume) was assessed by a modified version of the commonly used short version of the International Physical Activity Questionnaire [37,38], which focuses on intensity (nominally estimated in Metabolic Equivalents (MET) according to the type of activity) and duration (in minutes) of physical activity. We considered the following levels: brisk walking (≈3.3 METs), other activities of moderate intensity (≈4.0 METs) and activities of vigorous intensity (≈8.0 METs). In accordance with the current guidelines [39,40,41], these levels were used to assess adherence to guideline weekly exercise volume, using the following equations:Moderate intensity [MET·minutes/week] = (3.3 × minutes of brisk walking × days of brisk walking) + (4.0 × minutes of other moderate intensity activity × days of other moderate intensity activities)
Vigorous intensity: [MET·minutes/week] = 8.0 × minutes of vigorous intensity activity × days of vigorous intensity activity.
Total weekly physical activity volume [MET·minutes/week] = sum of Moderate + Vigorous MET·minutes/week scores

Our study population was subdivided into two groups: those (*n* = 711) reaching the physical activity goals as suggested by the latest guidelines [39,40,41], corresponding to at least to 150 min/week of moderate activity, or 75 min/week of vigorous activity, or a combination of both (above 600 [MET·minutes/week] considering total weekly physical activity volume), and those (*n* = 594) who are not reaching the physical activity goals (Below 600 [MET·minutes/week] considering total weekly physical activity volume).

Nutrition was assessed using the American Heart Association (AHA) Diet Score [23], considering fruit/vegetables, fish, sweetened beverages, whole-grain and sodium consumption (the assessment of the latter was adapted to Italian eating habits) [25].

Perception of stress, fatigue and somatic symptoms (short 4SQ) were assessed using a self-administered questionnaire [35,42,43], providing nominal self-rated Likert scales from 0 (‘no perception’) to 10 (‘highest perception’) for each measure. Short 4SQ considers four somatic symptoms; thus, the total score ranged from 0 to 40.

Smoke behavior: we considered all subjects who reported to have never smoked or to have stopped smoking more than one year ago as non-smokers.

To obtain a unique descriptor of lifestyle, as previously described [25], we considered three domains, nutrition (combination of AHA Diet Score and WC), exercise (total activity dose) and stress (combination of scores of somatic symptom, stress and fatigue perception). The three domains were combined into a single Index of Healthy Lifestyle, which ranged from 0 to 100 (with higher scores being healthier) using weights for measures of activity, diet, and stress according to our prior experience in a similar setting (see reference [25] for more details).

All participants voluntarily included anonymous data and they were aware of about the possible use of group data for scientific purpose.

### 2.2. Statistics

Summary data are presented as mean ± SD. The statistical strength of differences between groups was evaluated with GLM (General Linear Model) using gender and age as covariates. Simple correlations were used. Chi square tests were used for categorical variables. Computations were performed with a commercial statistical package (SPSS v27) (IBM, Armonk, NY, USA). A *p* < 0.05 was considered significant.

## 3. Results

Of the entire study population, 97% of workers are white collar, 1% blue collar and 2% students or retired employees. A total of 30% of employees were non-alcohol drinkers, 38% drank 1–3 glasses of wine/week, but no spirits, and 32% drank more than three glasses of wine/week or drank spirits. No significant differences were observed in all the reported data regarding the smoking and non-smoking population. Workers who reported suffering from chronic disease had a greater Body Mass Index (BMI), waist circumference and reported a lesser volume of vigorous physical activity. No significant differences were observed regarding other lifestyle determinants. A total of 81% of the employees who completed the questionnaire were employed in companies which benefited from Assidim services and were characterized by a higher Lifestyle Index as compared to employees who were employed in other companies (55.83 ± 20.6 vs. 50.87 ± 20.7 au, *p* = 0.046).

Table 1 reports the data of all subjects, together and subdivided by gender. Females were significantly younger than men, reported lower levels of strenuous physical activity and had higher scores of somatic symptoms, stress and fatigue perception. Moreover they presented a lower (unhealthy) Lifestyle Index value, including its exercise and stress determinants.

Table 2 reports the data regarding employees who reached the physical activity goals (Above 600 (MET·minutes/week) considering Total Weekly physical activity volume), and those who did not. More active employees presented with a lower BMI and waist circumference, reduced scores for somatic symptoms, stress and fatigue perception, and a better AHA Diet Score. The global lifestyle index was higher (healthier) in this population, particularly its stress component (see Figure 1). It is important to consider that the indexes, including the Stress Index, range from 0 to 100, and were built so that higher scores are healthier. More active employees report a higher perception of sleep quality, health quality and of job performance.

Table 3 report a simple correlation matrix between self-reported measures of nutrition, WC, somatic symptoms, stress and fatigue perception, sleep, and total volume of physical activity. Note the significant correlations between data, with the exception of sleep hours and AHA score.

## 4. Discussion

In this paper, we show the evidence for a better psychological and nutritional profile in the working population meeting the current physical activity recommendations, using a simple anonymous on-line questionnaire. We show also that active employees are characterized by a perception of better job performance, sleep and health quality.

Workplace Health Promotion (WHP) may represent an important sustainable tool for employers, employees and the global society. To this end, it is important that all companies, independently of their size, have WHP policies in place, offering to their employees (and their families) resources to improve lifestyle [3,4,5,6]. In this study, we verify the possibility of using a web-based questionnaire to improve employees’ awareness of their own lifestyle and companies’ awareness of grouped data, to foster an improvement in behaviors which can influence health, such as exercise, nutrition, smoking and sleep habits.

The possibility of collecting and managing anonymous data from several small companies also allows for the creation of useful feedback on their employees’ lifestyle (grouped with data of other similar small companies), while respecting the privacy of all employees, creating a benchmark population that reflects the characteristics and needs of similar companies, characterized by a similar socioeconomic status, and similar cultural and geographic backgrounds.

Awareness of one’s own lifestyle and health is an important initial step when fostering changes in behavior [8,18,20,26], and lifestyle management programs are more effective if tailored to an individual and/or company’s real situation, characteristics and preferences [8,10,11,17,18,19].

The questionnaire was hosted, as a part of an ongoing initiative to promote healthy lifestyle, on the website of a non-profit association (which provides associated companies, their employees and their families with financial assistance and support in case of disease, accident, invalidity and death, and which promotes a healthy lifestyle among workers and associated companies as its mission), which is to assist companies of any size, including small and medium-sized companies, which often do not have the resources required to introduce WHP policies without its support. In this study, we observed that employees who were employed in companies which benefit from Assidim services were characterized by a higher Lifestyle Index as compared to employees who were employed in other companies (55.83 ± 20.6 vs. 50.87 ± 20.7 au, *p* = 0.046). These data might be difficult to interpret, and might suggest both that Assidim-associated companies are more prone to offering health assistance to their employees, and that the programs offered by the non-profit association actually may help to foster good health. In a previous paper [25], we showed that, in an Italian multinational company, which offered an effective WHP program to its employees [44], the Lifestyle Index was 68.9 ± 20.6 au, and that this index was significantly related to key biochemical, hematological and hemodynamic variables predicting levels of cardiometabolic risk.

In this study, we observed that the perception of stress, fatigue and somatic symptoms related to stress (short 4SQ) were higher in women, and that the stress index (built combining these three domains) was lower (i.e., less healthy) than in men. Gender differences in perception of stress [45,46], are already known, and may be due to both genetic and social characteristics. The possibility of easily quantifying stress perception, using three questions about stress from a cognitive (directly asking about stress perception: “do you feel stressed” [47]) and a somatic (asking questions regarding perception of fatigue and other somatic symptoms, such as palpitations or muscular tension) perspective may offer a simple metric for WHP interventions [28,35,42,48]. Of particular interest is the role of physical activity as tool to manage stress [32,49,50,51], and in general as a tool to improve wellbeing and productivity [5,6,49,52]. In this study, we observed that physically active employees show a reduced perception of stress, associated with a better perception of own job performance, health and sleep quality. The mechanisms which may explain the role of exercise as a stress management tool are various and complex, and may include psychological effects; for instance, physical activity promotes positive changes in one’s mental health and ability to cope with stress [30], as well as physiological ones. In this regard, it is important to emphasize that aerobic endurance training may directly improve physiological control mechanisms, such as immunological, hormonal and autonomic nervous system controls [20,51], which are impaired in stress conditions [20,35,43,48]. Although stress could impair efforts to be physically active [29], exercise is considered a valid therapeutic strategy, and is employed in patients with depression and anxiety [53,54,55]. Notably, exercise, and a generally healthy lifestyle, are associated with better job performance [5,6,49,52], and the adoption of a WHP policy may improve productivity [3,4,5,6]. In this study, active employees report better perceived job performance. Perceived quality of health and sleep were also better in this active population, showing that this simple questionnaire is can also show important link between healthy lifestyle, performance and sleep quality, an important health determinant [56,57].

In this study, we evidence that active employees also present a better quality of nutrition, as indicated by the higher AHA diet Score, together with a reduced waist circumference as compared to non-active employees; furthermore, the nutrition index (built combining AHA diet score and WC) was higher.

Diet and exercise habits are often correlated [21,22,24], and subjects prone to exercise are characterized by a healthier diet (or vice versa). The important links between different lifestyle habits are also evident in this study: we observed significant correlations between total volume of physical activity, AHA diet score, WC, perception of stress, of fatigue of somatic symptoms and sleep hours.

Smoking, and the report on the presence of chronic disease, also need to be considered. In this study, we did not observe any significant differences when subdividing the study population into smoking employees and non-smoking ones. Employees who reported suffering from chronic disease were characterized by a higher BMI and WC and a reduced volume of vigorous exercise; no differences were observed regarding perception of stress, quality of diet, volume of moderate exercise or sleep hours. We have to consider that the questionnaire may not differentiate between chronic diseases (such as diabetes, hypertension, coronary artery disease) which are frequently associated with unhealthy lifestyle [21] from other chronic diseases which are not.

The data obtained by self-reported questionnaires might be of suboptimal quality. However, the high number of respondents and the quality analysis of the data may help in controlling this aspect. Moreover, the questionnaire was completely anonymous and we provided participants with a personalized immediate report based on the filled information, increasing their compliance [42] with the insertion of real data in order to obtain a report that actually referred to their condition. In this study, 84% of the employees reported a desire for more counseling to improve their lifestyle, showing a positive attitude towards information regarding their health status.

## 5. Conclusions

In conclusion, this study shows a clear better psychological and nutritional pattern, associated with the perception of better job performance and quality of health and sleep, in employees who meet the current physical activity recommendations, employed also in small–medium companies. These data may be useful when designing tailored WHP interventions in companies which may not be able to afford all the costs related to WHP, with the possibility of creating a metric regarding the intervention program that will be put in place. This possibility may help in the creation of effective programs and in improving employees’ and companies’ motivation and attitude towards WHP. In fact, as was also underlined by the Centers for Disease Control and Prevention (CDC) [58], WHP is more likely to be successful if they tailored to a company’s characteristics, and coordinated, planned or integrated to reduce health threats to workers both in and out of work, considering occupational safety and health in their design and execution [59,60]. To this end, a systematic process is welcomed when building a workplace health promotion program, considering specific steps such as a workplace health assessment, planning the program, implementing the program and determining its impact through evaluation [58]. Small and medium-size companies may encounter economic and cultural barriers when designing and implementing successful programs [61,62], and the number of companies which adopt WHP is low [63]. This latter consideration may have policy implications considering the high number of such companies and the relevant impact on the development of the world of primary and secondary prevention of chronic diseases. Policies might also consider the possibility of defining specific preventive methodologies targeted to small and medium-size companies [64], helping to create a specific wellness culture [44,65,66], which may have a positive impact outside the work environment. The data showing that small-size companies seem to achieve higher employee participation rates and more health improvements compared to larger companies are also of interest [67], moving from the view of WHP as a “luxury” to the perception of the high effectiveness of the program after its application [68]. Policies might consider tailored low-cost programs, re-orientation of work practices, tax incentives, and management support, so that the proportion of small–medium companies which adhere to WHP initiatives could increase [68]. The importance of including companies of any size in WHP is corroborated by the economic issues, considering the high relationship between economics, health and sustainability in the modern context [69]. Workplace health programs do not only impact health care costs in large companies with more than 1000 workers [4]; they also impact costs in smaller companies, as shown by a review of 73 published studies on worksite health promotion programs [70]. An investment in employee health may lower health care costs and insurance claims in employees who present a high cardio-metabolic risk (such as employees who do not meet current physical activity recommendations), improving their health, and in employees characterized by a low risk (such as physically active employees) by promoting health maintenance. A systematic review showed that well-implemented WHP can lead to 25% savings each on health care costs, absenteeism and workers’ compensation and disability management claims costs [71]. Moreover, individual employees can also save money by improving their health [58].

## Figures and Tables

**Figure 1 ijerph-18-08991-f001:**
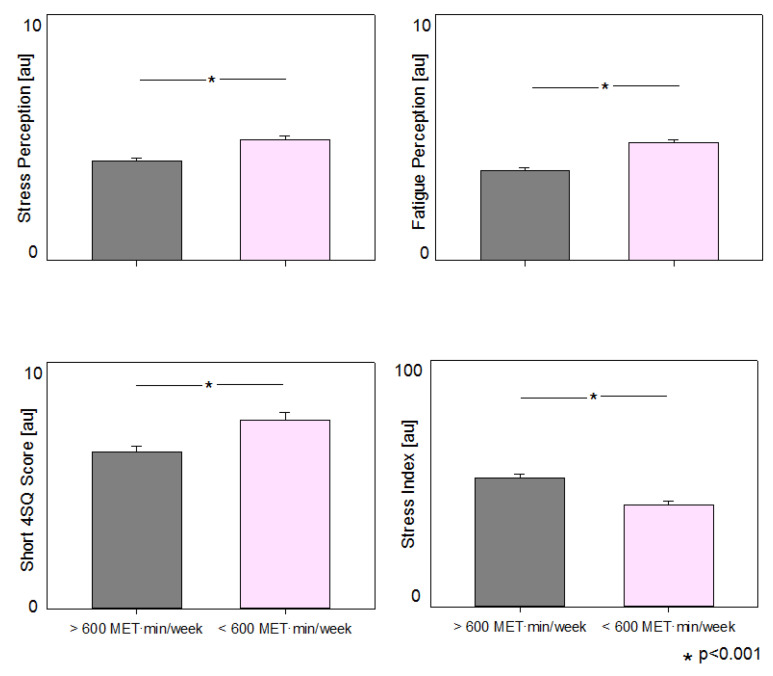
Perception of Stress, Fatigue, Subjective Stress-Related Somatic Symptoms (4SQ) and Stress Index in employees meeting current physical activity recommendations (Above 600 [MET·minutes/week) (**left** bars) and those who did not (Below 600 [MET·minutes/week) (**right** bars). Note that Stress Index (combination of scores of somatic symptoms, stress and fatigue perception) was built so that higher scores being healthier.

**Table 1 ijerph-18-08991-t001:** Anthropometric and lifestyle data collected in all subjects (total) and in female and male subgroups.

Variables	Total	Female	Male	Significance *p*
N	1305	620	685	
Age [yrs]	48.70 ± 11.21	45.63 ± 10.19	51.48 ± 11.37	<0.001
Weight [Kg]	71.64 ± 14.88	61.84 ± 11.51	80.48 ± 11.68	<0.001
BMI [Kg/m^2^]	24.12 ± 3.87	22.69 ± 4.02	25.41 ± 3.23	<0.001
Height [cm]	171.80 ± 9.10	165.06 ± 6.28	177.88 ± 6.59	<0.001
Waist circumference [cm]	86.89 ± 13.79	80.08 ± 12.59	94.07 ± 11.12	<0.001
Activity volume (moderate brisk walking) [MET·min/week]	394.71 ± 470.76	384.65 ± 470.49	404.87 ± 471.43	ns
Activity volume (other moderate activities) [MET·min/week]	265.32 ± 396.36	265.50 ± 401.98	265.59 ± 391.95	ns
Activity volume (vigorous) [MET·min/week]	401.30 ± 788.72	295.10 ± 692.71	498.49 ± 856.42	<0.001
Total Activity volume [MET·min/week]	1061.34 ± 1255	945.25 ± 1233	1168.94 ± 1268	0.01
AHA Diet Score [au]	2.22 ± 1.03	2.34 ± 1.06	2.12 ± 0.99	ns
short 4SQ score [au]	6.95 ± 7.30	8.57 ± 7.71	5.49 ± 6.59	<0.001
Smoke [*n* (%)]	186 (14.25)	92 (14.8)	94 (13.7)	ns
STRESS perception [au]	4.45 ± 3.00	5.15 ± 2.99	3.81 ± 2.87	<0.001
FATIGUE perception [au]	4.17 ± 2.96	4.93 ± 2.99	3.47 ± 2.74	<0.001
SLEEP [hours per night]	6.75 ± 1.09	6.82 ± 1.13	6.69 ± 1.06	ns
Perception of sleep quality [au]	6.26 ± 2.14	6.22 ± 2.18	6.29 ± 2.09	ns
Perception of HEALTH quality [au]	6.94 ± 1.57	6.90 ± 1.63	6.97 ± 1.52	ns
Perception of JOB PERFORMANCE [au]	4.26 ± 0.78	4.26 ± 0.74	4.25 ± 0.81	ns
NUTRITION index [au]	50.51 ± 11.52	51.25 ± 12.17	49.74 ± 10.77	ns
EXERCISE index [au]	67.71 ± 41.24	64.35 ± 42.10	70.87 ± 40.20	0.029
STRESS index [au]	47.42 ± 34.88	38.55 ± 22.89	55.56 ± 33.79	<0.001
LIFESTYLE INDEX [au]	54.87 ± 20.69	51.63 ± 20.32	58.31 ± 20.50	<0.001

Data are presented as mean ± SD; significance, according to GLM with age as covariate, consider differences between female and male subjects. Abbreviations: *p* = significance; BMI = body mass index; AHA = American Heart Association; MET = Metabolic Equivalent; 4SQ = Subjective Somatic Stress Symptoms Questionnaire; au = arbitrary units.

**Table 2 ijerph-18-08991-t002:** Anthropometric and lifestyle data observed in workers whose weekly physical activity is above or below the current physical activity goals.

Variables	Above	Below	Significance *p*
N	711	594	
Age [yrs]	49.38 ± 11.33	48.89 ± 11.00	0.017
Weight [Kg]	71.06 ± 13.95	72.30 ± 15.87	<0.001
BMI [Kg/m^2^]	23.73 ± 3.53	24.58 ± 4.20	<0.001
Height [cm]	172.51 ± 8.95	170.92 ± 9.18	0.338
Waist circumference [cm]	85.52 ± 12.87	88.74 ± 14.64	<0.001
Activity volume (moderate brisk walking) [MET·min/week]	186.57 ± 159.38	39.83 ± 51.10	<0.001
Activity volume (other moderate activities) [MET·min/week]	615.68 ± 525.96	131.43 ± 107.61	<0.001
Activity volume (vigorous) [MET·min/week]	726.23 ± 953.72	13.59 ± 58.20	<0.001
Total Activity volume [MET·min/week]	1787.52 ± 1307.45	195.04 ± 197.34	<0.001
AHA Score [au]	2.58 ± 0.04	2.22 ± 0.04	<0.001
short 4SQ score [au]	6.36 ± 9.94	7.67 ± 7.67	0.018
Smoke [*n* (%)]	91 (12.8)	95 (16.0)	ns
STRESS perception [au]	4.04 ± 2.79	4.92 ± 3.16	<0.001
FATIGUE perception [au]	3.66 ± 2.79	4.77 ± 3.03	<0.001
SLEEP [hours per night]	6.78 ± 1.08	6.71 ± 1.1	ns
Perception of sleep quality [au]	6.44 ± 2.00	6.05 ± 2.26	0.001
Perception of HEALTH quality [au]	7.30 ± 1.30	6.50 ± 1.74	<0.001
Perception of JOB PERFORMANCE [au]	4.31 ± 0.73	4.19 ± 0.81	0.002
NUTRITION index [au]	52.94 ± 10.66	47.59 ± 11.84	<0.001
EXERCISE index [au]	97.80 ± 3.17	31.83 ± 36.73	<0.001
STRESS index [au]	52.47 ± 33.82	41.50 ± 35.20	<0.001
LIFESTYLE INDEX [au]	67.51 ± 12.46	39.84 ± 18.34	<0.001

Data are presented as mean ± SD; significance, according to GLM with gender and age as covariates, consider differences between volume of physical activity. Abbreviations: *p* = significance; BMI = body mass index; AHA = American Heart Association; MET = Metabolic Equivalent; 4SQ = Subjective Somatic Stress Symptoms Questionnaire; au = arbitrary units.

**Table 3 ijerph-18-08991-t003:** Spearman’s Correlation within selected variables.

	AHA Diet Score	WC	Short 4SQ Score	STRESS Perception	FATIGUE Perception	Sleep Hours	Total Activity Volume [MET·min/week]
AHA Diet Score	1.000						
WC	−0.170 **	1.000					
	0.000						
Short 4SQ score	−0.091 **	−0.082 **	1.000				
	0.001	0.006					
STRESS perception	−0.126 **	−0.079 **	0.528 **	1.000			
	0.000	0.008	0.000				
FATIGUE perception	−0.136 **	−0.087 **	0.553 **	0.742 **	1.000		
	0.000	0.003	0.000	0.000			
Sleep hours	0.023	−0.099 **	−0.071 *	−0.125 **	−0.104 **	1.000	
	0.406	0.001	0.01	0.000	0.000		
Total Activity volume [MET·min/week]	0.186 **	−0.150 **	−0.115 **	−0.190 **	−0.221 **	0.057 *	1.000
	0.000	0.000	0.000	0.000	0.000	0.038	

** Correlation is significant at the 0.01 level (2-tailed). * Correlation is significant at the 0.05 level (2-tailed). WC = Waist circumference; 4SQ = Subjective Somatic Stress Symptoms Questionnaire; AHA American Heart Association; MET = Metabolic Equivalent.

## Data Availability

Data are available pending agreement with the Authors.

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
