# Peer review of "Evidence of Better Psychological Profile in Working Population Meeting Current Physical Activity Recommendations"

_ijerph, 2021, doi:10.3390/ijerph18178991_

Round 1
Reviewer 1 Report
This is a very good and a very interesting paper, well worth publishing. There are only two minor shortcomings: 1) the conclusion is very short, not to say minimal; I admit the discussion is quite substantial, but such a solid paper would deserve a somewhat sturdier conclusion; 2) there are many minor language mistakes (e.g. missing plural s, missing "a", etc.) too numerous to list here, so a thorough linguistic re-read would be appropriate.
Reviewer 2 Report
It is difficult to design tailored (and thus effective) WHP programs particularly in small-medium companies which rarely have access to sufficient economic and organizational resources. Therefore, this study is of great significance, especially for the development of small and medium-sized enterprises.However, there are still several issues to be confirmed before publication:
1.The conclusion needs to add policy implications, especially the more universal and effective promotion methods, which is of great significance to the development of the world.
2.Please check the whole paper, make sure the language fulfill the requirement of the journal.
3.In this study 1305 are employees randomly distributed? This is very important for the selection of samples and will directly affect the following conclusions. Data sources and selection criteria should be explained and confirmed in detail.
4.
The display of the research results of the article is like some research reports or statistical reports. In fact, academic papers focus on analyzing the internal mechanism behind them, which is very important for mastering the laws of economic management activities, so we should analyze the economic meaning or management enlightenment behind the data results.
5.The literature review part is too weak, the author has not given a relatively comprehensive review on the existing literature, and therefore it is hard to highlight the contribution of this research comparing with existing research.
Some related paper should be added, eg:
Sun H., Bless Kofi E., Sun C., Kporsu A K. Institutional quality, green innovation and energy efficiency, Energy policy, (2019),135,111002. https://doi.org/10.1016/j.enpol.2019.111002.
Round 2
Reviewer 2 Report
The article has become a good model after modification. At present, the data are detailed and the demonstration is powerful. Therefore, I propose to publish it directly.